# Improving Natural Language Processing Tasks with Human Gaze-Guided Neural Attention

**Ekta Sood**[1], **Simon Tannert**[2], **Philipp Müller**[1], **Andreas Bulling**[1]
[1]University of Stuttgart, Institute for Visualization and Interactive Systems (VIS), Germany
[2]University of Stuttgart, Institute for Natural Language Processing (IMS), Germany
`{ekta.sood, philipp.mueller, andreas.bulling}@vis.uni-stuttgart.de`
`simon.tannert@ims.uni-stuttgart.de`

## Abstract

A lack of corpora has so far limited advances in integrating human gaze data as a supervisory signal in neural attention mechanisms for natural language processing (NLP). We propose a novel *hybrid text saliency model* (TSM) that, for the first time, combines a cognitive model of reading with explicit human gaze supervision in a single machine learning framework. On four different corpora we demonstrate that our hybrid TSM duration predictions are highly correlated with human gaze ground truth. We further propose a novel *joint modeling approach* to integrate TSM predictions into the attention layer of a network designed for a specific upstream NLP task without the need for any task-specific human gaze data. We demonstrate that our joint model outperforms the state of the art in paraphrase generation on the Quora Question Pairs corpus by more than 10% in BLEU-4 and achieves state of the art performance for sentence compression on the challenging Google Sentence Compression corpus. As such, our work introduces a practical approach for bridging between data-driven and cognitive models and demonstrates a new way to integrate human gaze-guided neural attention into NLP tasks.

## 1   Introduction

Neural attention mechanisms have been widely applied in computer vision and have been shown to enable neural networks to only focus on those aspects of their input that are important for a given task [48, 81]. While neural networks are able to learn meaningful attention mechanisms using only supervision received for the target task, the addition of human gaze information has been shown to be beneficial in many cases [32, 58, 80, 84]. An especially interesting way of leveraging gaze information was demonstrated by works incorporating human gaze into neural attention mechanisms, for example for image and video captioning [71, 83] or visual question answering [58].

While attention is at least as important for reading text as it is for viewing images [13, 78], integration of human gaze into neural attention mechanisms for natural language processing (NLP) tasks remains under-explored. A major obstacle to studying such integration is data scarcity: Existing corpora of human gaze during reading consist of too few samples to provide effective supervision for modern data-intensive architectures and human gaze data is only available for a small number of NLP tasks. For paraphrase generation and sentence compression, which play an important role for tasks such as reading comprehension systems [23, 28, 54], no human gaze data is available.

We address this data scarcity in two novel ways: First, to overcome the low number of human gaze samples for reading, we propose a novel hybrid text saliency model (TSM) in which we combine a cognitive model of reading behavior with human gaze supervision in a single machine learning framework. More specifically, we use the E-Z Reader model of attention allocation during reading [60] to obtain a large number of synthetic training examples. We use these examples to

pre-train a BiLSTM [22] network with a Transformer [75] whose weights we subsequently refine by training on only a small amount of human gaze data. We demonstrate that our model yields predictions that are well-correlated with human gaze on out-of-domain data. Second, we propose a novel joint modeling approach of attention and comprehension that allows human gaze predictions to be flexibly adapted to different NLP tasks by integrating TSM predictions into an attention layer. By jointly training the TSM with a task-specific network, the saliency predictions are adapted to this upstream task without the need for explicit supervision using real gaze data. Using this approach, we outperform the state of the art in paraphrase generation on the Quora Question Pairs corpus by more than 10% in BLEU-4 and achieve state of the art performance on the Google Sentence Compression corpus. As such, our work demonstrates the significant potential of combining cognitive and data-driven models and establishes a general principle for flexible gaze integration into NLP that has the potential to also benefit tasks beyond paraphrase generation and sentence compression.

## 2 Related work

Our work is related to previous works on 1) NLP tasks for text comprehension, 2) human attention modeling, as well as 3) gaze integration in neural network architectures.

### 2.1 NLP tasks for text comprehension

Two key tasks in machine text comprehension are paraphrasing and summarization [8, 28, 9, 41, 24]. While paraphrasing is the task of "conveying the same meaning, but with different expressions" [9, 18, 41], summarization deals with extracting or abstracting the key points of a larger input sequence [21, 73, 33]. Though advances have helped bring machine comprehension closer to human performance, humans are still superior for most tasks [3, 79, 85]. While attention mechanisms can improve performance by helping models to focus on relevant parts of the input [57, 65, 63, 7, 27, 11], the benefit of explicit supervision through human attention remains under-explored.

### 2.2 Human attention modeling

Predicting what people visually attend to in images (saliency prediction) is a long-standing challenge in neuroscience and computer vision [4, 6, 40]. In contrast to images, most attention models for eye movement behaviors during reading are cognitive process models, i.e. models that do not involve machine learning but implement cognitive theories [17, 59, 60]. Key challenges for such models are a limited number of parameters, hand-crafted rules and thus a difficulty to adapt them to different tasks and domains, as well as the difficulty to use them as part of an end-to-end trained machine learning architectures [16, 39, 45]. One of the most influential cognitive models of gaze during reading is the E-Z Reader model [60] It assumes attention shifts to be strictly serial in nature and that saccade production depends on different stages of lexical processing. that has been successful in explaining different effects seen in attention allocation during reading [61, 62].

In contrast, learning-based attention models for text remain under-explored. Nilsson and Nivre [50] trained person-specific models on features including length and frequency of words to predict fixations and later extended their approach to also predict fixation durations [51]. The first work to present a person-independent model for fixation prediction on text used a linear CRF model [46]. A separate line of work has instead tried to incorporate assumptions about the human reading process into the model design. For example, the Neural Attention Trade-off (NEAT) language model was trained with hard attention and assigned a cost to each fixation Hahn and Keller [25]. Subsequent work applied the NEAT model to question answering tasks, showing task-specific effects on learned attention patterns that reflect human behavior [26]. Further approaches include sentence representation learning using surprisal and part of speech tags as proxies to human attention [76], attention as a way to improve time complexity for NLP tasks [68], and learning saliency scores by training for sentence comparison [66]. Our work is fundamentally different from all of these works in that we, for the first time, combine cognitive theory and data-driven approaches.

### 2.3 Gaze integration in neural network architectures

Integration of human gaze data into neural network architectures has been explored for a range of computer vision tasks [32, 69, 80, 83, 84]. Sugano and Bulling [71] were the first to use gaze as an

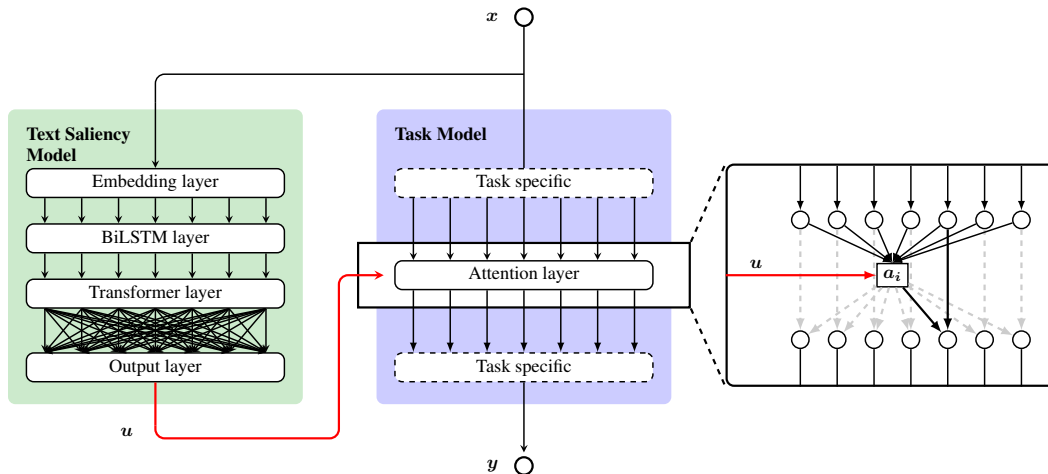

Figure 1: High-level architecture of our model. Given an input sentence $x_1...x_n$ the Text Saliency Model produces attention scores $u_1...u_n$ for each word in the input sentence $x$. The Task Model combines this information with the original input sentence to produce an output sentence $y_1...y_m$.

additional input to the attention layer for image captioning, while Qiao et al. [58] used human-like attention maps as an additional supervision for the attention layer for a visual question answering task. Most previous work in gaze-supported NLP has used gaze as an input feature, e.g. for syntactic sequence labeling [36], classifying referential versus non-referential use of pronouns [82], reference resolution [30], key phrase extraction [86], or prediction of multi-word expressions [64]. Recently, Hollenstein et al. [29] proposed to build a lexicon of gaze features given word types, overcoming the need for gaze data at test time. Two recent works proposed methods inspired by multi-task learning to integrate gaze into NLP classification tasks. [37] did not integrate gaze into the attention layers but demonstrated performance improvements by adding a gaze prediction task to regularize a sentence compression model. [2] did not predict human gaze for the target task but used ground-truth gaze from another eye tracking corpus to regularize their neural attention function. In stark contrast, our work is the first to combine a cognitive model of reading and a data-driven approach to predict human gaze, to directly integrate these predictions into the neural attention layers, and to jointly train for two different tasks – generative (paraphrase generation) and classification (sentence compression).

## 3   Method

We make two contributions: A hybrid text saliency model as well as two attention-based models for paraphrase generation and text summarization employed in a novel joint modeling approach[1].

### 3.1   Hybrid text saliency model

To overcome the limited amount of eye-tracking data for reading comprehension tasks, we propose *a hybrid approach* when training our text saliency model. In the first stage of training, we leverage the E-Z Reader model [60] to generate a large amount of training data over the CNN and Daily Mail Reading Comprehension Corpus [28]. After training the text saliency model until convergence using this synthetic data, in a second training phase we fine-tune the network with real eye tracking data of humans reading from the Provo and Geco corpus [43, 14]. We used the most recent implementation of EZ Reader (Version 10.2) available from the authors' website[2].

The task of text saliency is to predict fixation durations $u_i$ for each word $x_i$ of an input sentence. In our text saliency model, we combine a BiLSTM network [22] with a Transformer [75] (see Figure 1 for an overview). Each word $x_i$ of the input sentence is encoded using pre-trained GloVe

embeddings [55]. The resulting embeddings are fed into a single-layer BiLSTM network [22] that integrates information over the whole input sentence. The outputs from the BiLSTM network are fed into a Transformer network with multi-headed self-attention [75]. In contrast to [75], we only use the encoder of the Transformer network. Furthermore, we do not provide positional encodings as input because this information is already implicitly present in the outputs of the BiLSTM layer. In preliminary experiments we found advantages in using only four layers with four attention heads each for the Transformer network in contrast to the six layers with 12 heads in the original architecture [75]. We also found the combination of a BiLSTM network with a subsequent Transformer network to yield predictions that are most similar to human data, particularly with longer sequence lengths. According to [77], this might be due to the transformer encoding coarse relational information about positions of sequence elements, while the BiLSTM better captures fine-grained word level context. The specific choice of architecture therefore allows our model to better capture the sequential context while still maintaining computational efficiency. Finally, a fully connected layer is used to obtain an attention score $u_i$ for each input word $x_i$ in $x$. We apply sigmoid nonlinearities with subsequent normalization over the input sentence to obtain a probability distribution over the sentence. As loss function we use the mean squared error.

## 3.2 Joint modeling for natural language processing tasks

To model the relationship between attention allocation and text comprehension, we integrate the TSM with two different NLP task attention-based networks in a *joint model* (see Figure 1). Specifically, we propose a modification to the Luong attention layer [44] that is a computationally light-weight but highly effective, multiplicative attention algorithm [44, 5]. We compute attention scores $a_i$ as

$$a_i = \text{softmax}(\text{score}_\text{T}(h_i, s_j)) \tag{1}$$

using our task-specific modified score functions $\text{score}_\text{T}$. For the tasks of paraphrase generation and sentence compression, respectively, we propose the novel score functions

$$\text{score}_\text{ParaGen}(h_i, s_j) = u \odot h_i^\top W_a s_j \tag{2}$$

$$\text{score}_\text{TextComp}(h_i, s_j) = u \odot v_a^\top \tanh(W_a[h_i; s_j]) \tag{3}$$

Where $h_i$ is the current hidden state, $s_j$ are the hidden states of the encoder and $W_a$ and $v_a$ are learnable parameters of the attention mechanism. The outputs of the TSM model $u$ on the input sentence are incorporated into the score function by element-wise multiplication. This way, attention scores in the upstream task network reflect word saliencies learnt from humans. In addition to that, the error signal from the upstream loss function can be propagated back to the TSM in order to adapt its' parameters to the upstream task, thereby defining an implicit loss on $u$. This way, the attention distribution $u$ returned by the TSM is adapted to the specific upstream task, allowing us to incorporate and adapt a neural model of attention to tasks for which no human gaze data is available. Note, as we have two different tasks namely generative (paraphrase generation) and classification (sentence compression), we used different score functions as suggested by previous work [44].

## 4 Experiments

### 4.1 Joint model with upstream tasks

**Evaluation details**

**Datasets** We used two standard benchmark corpora to evaluate each upstream NLP task. For paraphrase generation, we used the Quora Question Pairs corpus[3] that consists of human-annotated pairs of paraphrased questions that were crawled from Quora. We followed the common practice of excluding negative paraphrase examples from the corpus to obtain training data for paraphrase generation [54, 23]. We split the data according to [23, 54], using either 100K or 50K examples for training, 45K examples for validation, and 4K examples for testing. For the sentence compression

Table 1: Ablation study results and comparison with the state of the art for paraphrase generation with both data splits in terms of BLEU-4 score for different training set sizes and sentence compression in terms of F1 score and compression ratio. Also shown is the number of model parameters.

| Paraphrase Generation (BLEU-4) | | | | Sentence Compression | | | |
|---|---|---|---|---|---|---|---|
| Method | 50K | 100K | Params | Method | F1 | CR | Params |
| | | | | Klerke et al. (2016) | 80.9 | — | — |
| Baseline (Seq-to-Seq) | 7.11 | 8.91 | 45M | Baseline (BiLSTM) | 81.3 | 0.39 | 12M |
| Patro et al. (2018) | 16.5 | 17.9 | — | Zhao et al. (2018) | **85.1** | 0.39 | — |
| No Fixation | 24.62 | 27.81 | 69M | No Fixation | 83.4 | 0.38 | 129M |
| Random TSM Init | 25.26 | 27.11 | 79M | Random TSM Init | 83.7 | 0.38 | 178M |
| TSM Weight Swap | 23.43 | 27.60 | 79M | TSM Weight Swap | 83.8 | 0.38 | 178M |
| Frozen TSM | 25.73 | 28.26 | 79M | Frozen TSM | 83.9 | 0.37 | 178M |
| Ours | **26.24** | **28.82** | 79M | Ours | **85.0** | 0.39 | 178M |

task we used the Google Sentence Compression corpus [20] containing 200K sentence compression pairs that were crawled from news articles. We split the data according to [87], taking the first 1K examples as test data, and the next 1K as validation data.

**Paraphrase generation**    Our first text comprehension task was paraphrase generation where, given a source sentence, the model has to produce a different target sentence with the same meaning that may have a different length. We used a sequence-to-sequence network with word-level attention that was originally proposed for neural machine translation [1]. The model consisted of two recurrent neural networks, an encoder and an attention decoder (see Figure 1). The encoder consisted of an embedding layer followed by a gated recurrent unit (GRU) [10]. The decoder produced an output sentence step-by-step given the hidden state of the encoder and the input sentence. At each output step, the encoded input word and the previous hidden state are used to produce attention weights using our modified Luong attention (see Equation 2). These attention weights are combined with the embedded input sentence and fed into a GRU to produce an output sentence. The loss between predicted and the ground-truth paraphrase was calculated over the entire vocabulary using cross-entropy.

**Sentence compression**    As a second task, we opted for deletion-based sentence compression that aims to delete unimportant words from an input sentence [31, 38, 47, 12, 20]. We incorporated the attention mechanism into the baseline architecture presented in [20]. The network consisted of three stacked LSTM layers with dropout after each LSTM layer as a regularization method. The outputs of the last LSTM layer were fed through our modified Luong attention mechanism (see Equation 3) and two fully connected layers which predicted for each word whether it should be deleted. The loss between predicted and ground truth deletion mask was calculated with cross-entropy.

**Training**    We used pre-trained 300-dimensional GloVe embeddings in both the TSM and the upstream task network to represent the input words [55]. We trained both upstream task models using the ADAM optimizer [35] with a learning rate of 0.0001. For paraphrase generation we used uni-directional GRUs with hidden layer size 1,024 and dropout probability of 0.2. For sentence compression we used BiLSTMs with hidden layer size 1,024 and dropout probability of 0.1.

**Metrics**    The most common metric to evaluate text generative tasks is BLEU [53], which measures the n-gram overlap between the produced and target sequence. To ensure reproducibility, we followed the standard Sacrebleu [56] implementation that uses BLEU-4. For sentence compression, we followed previous works [20, 87] by reporting the F1 score as well as the compression ratio calculated as the length of the compressed sentence divided by the input sentence length measured in characters [20].

**Results and discussion**

Results for our joint model on paraphrase generation and sentence compression in comparison to the state of the art are shown in Table 1. As can be seen in the table, for paraphrase generation our approach achieves a BLEU-4 score of 28.82 when using 100K training examples, clearly

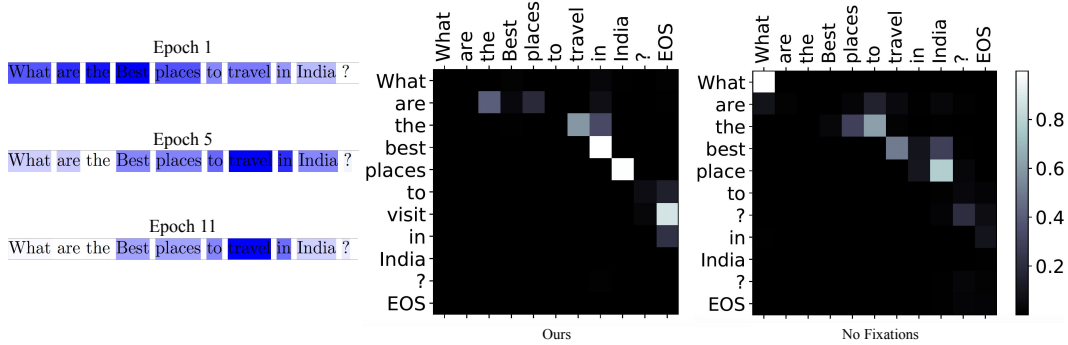

Figure 2: Paraphrase generation attention maps for both sub-networks (TSM predictions and upstream task attention) in our joint architecture. We show the TSM fixation predictions (left in blue) over epochs (last epoch is our converged models). We show the two-dimensional neural attention maps (right), showing our model and the *no fixation* model from our ablation study. The two-dimensional maps show the input sequence (horizontal axis) and the predicted sequence (vertical axis). We show the temporal TSM predictions over epochs, in order to depict how the fixation predictions change while training. The fixation predictions (for each epoch, left) are computed over words in the input sequences and then are integrated into the neural attention mechanism which in turn is used to make a prediction (vertical axis, right).

outperforming the previous state of the art for this task from [54] (17.9 BLEU-4). The same holds for 50K training examples (26.24 vs. 16.5 BLEU-4). For sentence compression, our joint model achieves a F1 score of 85.0 and a compression rate of 0.39. This is on par with the state of the art performance of 85.1 F1 score and 0.39 compression rate reported in Zhao et al. [87][4]. In that work, a syntax-based language model was used to learn the syntactic dependencies between lexical items in the given input sequence. In contrast, our current method does not require any syntax-based language model, but it will be interesting to see whether it will benefit from additional syntactic information in future work. When comparing our results for sentence compression on the Google dataset to [37] we observe an increase of ~5% F1 score for our method (cf. Table 1).

To further analyze the impact of our joint modeling approach, we evaluated several ablated versions of our model:

- **Baseline (Seq-to-Seq):** Stand-alone models based on a Seq-2-seq network [1] for paraphrase generation and a BiLSTM network [67] for sentence compression.

- **No Fixations:** Stand-alone upstream task network with original Luong attention (no TSM).

- **Random TSM Init:** Random initialization of the TSM instead of training on E-Z Reader and human data. Still implicit supervision by the upstream task during joint training.

- **TSM Weight Swap:** Exchange of the weights of the TSM model between tasks, i.e. sentence compression using the TSM weights obtained from the best-performing paraphrase generation model and vice versa.

- **Frozen TSM:** Training of the TSM with E-Z Reader and human gaze predictions but with frozen weights in the joint training with the upstream task, i.e. no adaptation of the TSM.

As can be seen from Table 1, all ablated models obtain inferior performance to our full model on both tasks (statistically significant at the 0.05 level). Notably, even the *No Fixation* model improves drastically over the *Seq-to-Seq baseline* for paraphrase generation, most likely due to the significant increase in network parameters. The benefit of training the *TSM* with our hybrid approach before using it in the joint model is underlined by the performance difference between the *Random TSM Init* (e.g. decrease in performance for both tasks) and our full model (e.g. best performance and differently adapted saliency predictions (see Table 1 and Figure 2).[5]

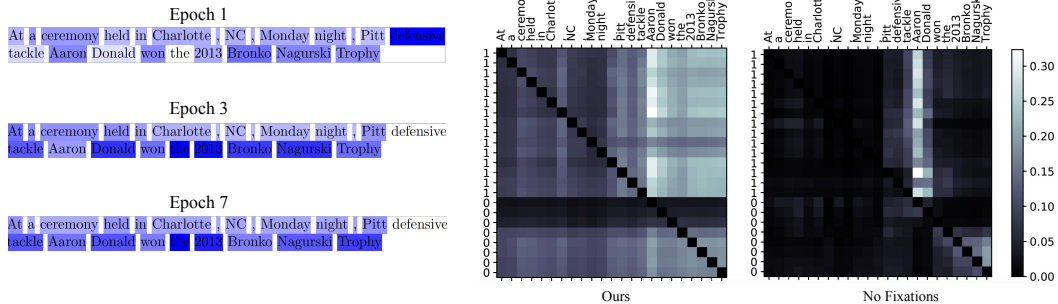

Figure 3: Sentence compression attention maps for both sub-networks (TSM predictions and upstream task attention) in our joint architecture. We show the TSM fixation predictions (left in blue) over epochs (last epoch is our converged models). We show the two-dimensional neural attention maps (right), showing our model and the *no fixation* model from our ablation study. The two-dimensional maps show the input sequence (horizontal axis) and the predicted sequence (vertical axis). We show the temporal TSM predictions over epochs, in order to depict how the fixation predictions change while training. The fixation predictions (for each epoch, left) are computed over words in the input sequences and then are integrated into the neural attention mechanism which in turn is used to make a prediction (vertical axis, right).

Most importantly, our full model achieves higher performance than the *Frozen TSM* model in all evaluations (e.g. 85.0 vs. 83.9 F1 for sentence compression), indicating that our model successfully adapts the TSM predictions during joint training. This is further underlined by the inferior performance of the *TSM Weight Swap* model: Swapping the optimal TSM weights between different upstream tasks leads to a notable performance decrease (e.g. 85.0 vs. 83.7 F1 for sentence compression), implying that the TSM model adaptation is specific to the upstream task.

To gain insights into how our joint model training adapts TSM predictions to specific upstream tasks, we analyzed the saliency predictions over time. Figure 2 shows a visualization of representative samples for both tasks over multiple training epochs. As can be seen in the left half of the figure, the adapted saliency predictions differ significantly from each other. In paraphrase generation (top) the saliency predictions focus on fewer words in the sentence within 11 epochs, specifically the word "travel" that is replaced in the correct paraphrase by "visit". For sentence compression (bottom) the predictions continue to be spread over the whole sentence with only slight changes in the distribution over the words. This makes sense given that the task of this network is to delete as many words in the input sequence as possible while still maintaining syntactic structure and meaning.

The right half of the figure shows 2D neural attention maps of the converged models with the input sequence on the horizontal and the prediction on the vertical axis for *our* (with fixations) and the *No Fixation* model, respectively. As can be seen, our model correctly predicts the paraphrase, while the *No Fixation* model does not. Also, both the converged models neural attention weights differ with respect to allocation of probability mass. We see the *No Fixation* model densely concentrates attention towards a specific few input words (horizontal axis) when predicting several words (vertical axis). In contrast, the attention mass of our model is more spread out.

## 4.2 Pre-training of the hybrid text saliency model (TSM)

### Evaluation details

**Training datasets** Training the TSM consists of two stages: pre-training with synthetic data generated by E-Z Reader, and subsequent fine-tuning on human gaze data. For the first step, we run E-Z Reader on the CNN and Daily Mail corpus [28] consisting of 300k online news articles with on average 3.75 sentences. As recommended in Reichle et al. [60], we run E-Z Reader 10 times for each sentence to ensure stability in fixation predictions. For training we obtain a total of 7.6M annotated sentences on Daily Mail and 3.1M for CNN. For validation, we obtained 850K sentences on Daily Mail and 350K on CNN. For the second step, we used the two established gaze corpora Provo [43] and Geco [14]. Provo contains 55 short passages, extracted from different sources such as popular

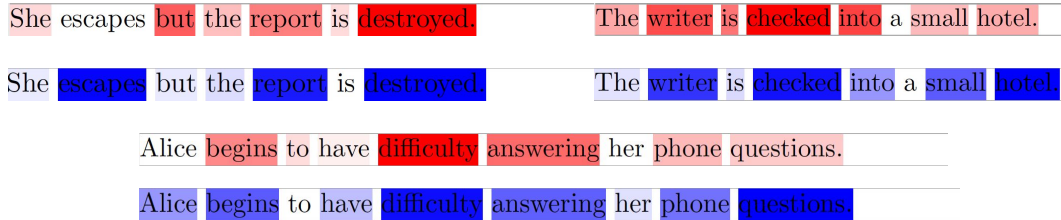

Figure 4: Heatmaps showing human fixation durations in red and hybrid TSM duration predictions in blue. Here we show three different example sentences in order to depict the similarity between TSM word-level durations predictions as compared to human ground truth word-level durations

Table 2: Comparison of predicted and human ground-truth fixation durations for the different TSM conditions and corpora in terms of mean squared error (MSE), Jensen Shannon Divergence (JSD), and Spearman's rank correlation ($\rho$) between the part of speech tags based fixation distributions for model predictions and ground truth. A star indicates statistically significant $\rho$ at $p < 0.05$.

| Corpus | TSM | | | TSM w/o pre-training | | | TSM w/o fine-tuning | | |
| --- | --- | --- | --- | --- | --- | --- | --- | --- | --- |
| | MSE | JSD | $\rho$ | MSE | JSD | $\rho$ | MSE | JSD | $\rho$ |
| Dundee | 0.063 | 0.39 | 0.99* | 0.071 | 0.39 | 0.99* | 0.096 | 0.47 | -0.68 |
| Provo + Geco | 0.105 | 0.34 | 1.00* | 0.112 | 0.36 | 0.99* | 0.238 | 0.46 | 0.10 |
| Provo | 0.003 | 0.24 | 0.88* | 0.008 | 0.44 | 0.83* | 0.032 | 0.52 | -0.25 |
| Geco | 0.118 | 0.35 | 0.99* | 0.127 | 0.35 | 0.98* | 0.267 | 0.45 | -0.10 |
| MQA-RC | 0.064 | 0.36 | 0.94* | 0.071 | 0.36 | 0.76* | 0.083 | 0.42 | -0.05 |

science magazines and fiction stories [43]. We split the data into 10K sentence pairs (pairs means sentence to human, as multiple humans read the same sentence) for train and 1K sentence pairs for validation. Geco is comprised of long passages from a popular novel [14]. We split the data into 65K sentence pairs for train and 8K sentence pairs for validation.

**Test datasets**    We evaluated our model on the validation sets of the Provo and Geco corpora, as well as on the Dundee [34] and MQA-RC corpora [70]. The combined validation corpora of Provo and Geco contained 18K sentence pairs. Dundee consists of recordings from 10 participants reading 20 news articles while MQA-RC corpus is a 3-condition reading comprehension corpus using 32 documents from the MovieQA question answering dataset [72]. For our evaluation we used 1K sentence pairs from the free reading condition. This dataset is substantially different from the other eye tracking corpora because its stimuli are scraped from online sources and contain noise not found in text intended for human reading.

**Implementation details**    We used pre-trained 300 dimensional GloVe word embeddings [55]. Our network has a bidirectional LSTM, with four transformer self-attention layers with four heads and hidden size of 128. The model objective is to predict normalized fixation durations for each word in the input sentence, resulting in saliency scores between 0 and 1. We used the ADAM optimizer [35] with a learning rate of 0.00001, batch size of 100, and dropout of 0.5 after the embedding layer and the recurrent layer. We pre-trained our network on synthetic training data for four epochs, and then fine-tune it on human data for 10 epochs.

**Metrics**    To evaluate the TSM model, we compute mean squared error (MSE) between the predicted and ground truth fixation durations as well as the Jensen-Shannon Divergence (JSD) [42]. JSD is widely used in eye tracking research to evaluate inter-gaze agreement [49, 19, 15, 52] as, unlike Kullback-Leibler Divergence, JSD is symmetric. In addition we measured the word type predictability as it is a well-known predictor of fixation probabilities [25, 50]. We used the Stanford tagger [74] to predict part-of-speech (POS) tags for our corpora and compute the average fixation probability per tag, allowing us to compute the correlation between our model and ground truth using Spearman's $\rho$.

**Results and discussion**

Table 2 shows the performance of our model and ablation conditions in terms of means squared error (MSE), Jensen-Shannon-Divergence (JSD) and correlation to human ground truth. As ablation conditions we evaluate a model only trained on human data (w/o pre-train) as well as a model that is not fine-tuned on human data (w/o fine-tune), but only trained with E-Z Reader data.

Most importantly, our model is superior to- or on par with both ablation variants across all metrics and corpora, showing the importance of both the E-Z Reader pre-training as well as the fine-tuning with human data. Pre-training with data obtained from E-Z Reader is most beneficial in the case of the small Provo corpus, where we observe a reduction from 0.44 JSD to 0.24 JSD by adding the pre-training step. For the larger corpora this difference is less pronounced but still present. It is interesting to note that TSM w/o fine-tune performs consistently the worst, indicating that training on E-Z Reader data alone insufficient even though it provides benefits when combined with human data.

Using the correlations to human gaze over the POS distributions, we can compare our approach to Hahn and Keller [25] who achieved a $\rho$ of 0.85 on the Dundee corpus, compared to a $\rho$ of 0.99 achieved by our model. Furthermore we observe an especially large improvement in $\rho$ as a result of E-Z Reader pre-training on the MQA-RC dataset. This dataset, unlike the other eye tracking corpora, is generated from stimuli which were scraped from online sources regarding movie plots, underlining the effectiveness of our approach in generalizing to out-of-domain data. In further analyses on the POS based correlations we observed that content words, such as adjectives, adverbs, nouns, and verbs, are more predictive than function words. [6] Lastly, we provide a qualitative impression of our method by comparing attention maps using our TSM predictions to ground truth human data (see Figure 4).

## 5 Conclusion

In this work we made two novel contributions towards improving natural language processing tasks using human gaze predictions as a supervisory signal. First, we introduced a novel hybrid text saliency model that, for the first time, integrates a cognitive reading model with a data-driven approach to address the scarcity of human gaze data on text. Second, we proposed a novel joint modeling approach that allows the TSM to be flexibly adapted to different NLP tasks without the need for task-specific ground truth human gaze data. We showed that both advances result in significant performance improvements over the state of the art in paraphrase generation as well as competitive performance for sentence compression but with a much less complex model than the state of the art. We further demonstrated that this approach is effective in yielding task-specific attention predictions. Taken together, our findings not only demonstrate the feasibility and significant potential of combining cognitive and data-driven models for NLP tasks – and potentially beyond – but also how saliency predictions can be effectively integrated into the attention layer of task-specific neural network architectures to improve performance.

## Broader Impact

We identified a number of potential benefits and risks of our approach.

**Potential benefits**   Our approach could benefit interactive applications that quantify, support, or enhance reading behavior without the need for special-purpose eye tracking equipment. Specifically, we see potential for e-learning applications in which our approach could be used to qualify reader actions and provide feedback to encourage improvement in reading comprehension. In addition, we see potential for our approach to be used as a key component in diagnostic tools to identify atypical eye movement behaviors correlated to cognitive impairments such as learning disabilities (e.g. such as Autism Spectrum Disorder or ADHD) or early onset detection of neuro-degenerative diseases which impact eye movement control (e.g Parkinsons disease). In addition, our hybrid approach could be useful for researchers building computational models of cognition, specifically geared towards combining traditional cognitive process models with neural networks in order to build a model which better emulates human cognitive processes – potentially allowing for increase in parameters and task complexity for further more robust models of human behavior. Lastly, our joint learning approach might prove useful to machine learning researchers who aim to implement artificial systems that more similarly model human behavior and thus perform more similarly to humans on currently challenging machine comprehension tasks. Our method could be used, for example, to automatically generate paraphrases for more natural playing experience in computer games (e.g., more varied utterances of non-player characters); or to aid in making a news text faster to read by compressing it before presenting it to the user.

**Potential risks**   Despite these potential future benefits, we also identified a few risks. Given the nature of the underlying E-Z reader model, it is likely that reading behavior found in atypical individuals currently cannot be predicted. This could consequently negatively impact the quality of the provided feedback, guidance or diagnostic analyses. We also see a potential risk in that our improved paraphrase generation model could be used to disguise plagiarized texts. Finally, improvements in sentence compression could make users grow accustomed to simplified and potentially less balanced information if they increasingly read compressed news articles.

## Acknowledgments and Disclosure of Funding

E. Sood was funded by the Deutsche Forschungsgemeinschaft (DFG, German Research Foundation) under Germany's Excellence Strategy - EXC 2075 – 390740016; S. Tannert was supported by IBM Research AI through the IBM AI Horizons Network; P. Müller and A. Bulling were funded by the European Research Council (ERC; grant agreement 801708). Additional revenues related to, but not supporting, this work: Scholarship by Google for E. Sood. We would like to thank the following people for their helpful insights and contributions: Sean Papay, Pavel Denisov, Prajit Dhar, Manuel Mager, Diego Frassinelli, Fabian Koegel and Keerthana Jaganathan.

## Footnotes

[1]Code and other supporting material can be found at `https://perceptualui.org/publications/sood20_neurips/`

[2]`http://www.erikdreichle.com/downloads.html`

[3]`https://www.quora.com/q/quoradata/First-Quora-Dataset-Release-Question-Pairs`

[4]For a more detailed comparison to our model see table 1 in the supplementary material.

[5]Additional 1D and 2D maps over all conditions are available in the supplementary material.

[6]Detailed POS distributions are available in the supplementary material.

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
