[Supplementary Material]

# Improving Natural Language Processing Tasks with Human Gaze-Guided Neural Attention: Supplementary Material

**Ekta Sood**[1], **Simon Tannert**[2], **Philipp Müller**[1], **Andreas Bulling**[1]

[1]University of Stuttgart, Institute for Visualization and Interactive Systems (VIS), Germany
[2]University of Stuttgart, Institute for Natural Language Processing (IMS), Germany

{sood, mueller, bulling}@vis.uni-stuttgart.de      simon.tannert@ims.uni-stuttgart.de

## 1 Sentence Compression Comparison To Previous SOTA

To gain further insight into the comparison between our model and the current state of the art in sentence compression, we show results of our method and ablations in relation to ablations of the method by Zhao et al. [4] (see Table 1). In their work, the authors added a "syntax-based language model" to their sentence compression network with which they obtained the state-of-the-art performance of 85.1 F1 score. The authors employ a syntax-based language model which is trained to learn the syntactic dependencies between lexical items in the given input sequence. Together with this language model, they use a reinforcement learning algorithm to improve the deletion proposed by their Bi-LSTM model. Using a naive language model without syntactic features their model obtained a F1 score of 85.0. With their stand-alone Bi-LSTM method in which they do not employ the reinforce language model policy, they obtain 84.8. In contrast, our method does neither include a reinforcement-learning based language model nor additional syntactic features. However, our method is still competitive with the state of the art (achieving a F1 score of 85.0), and arguably might benefit from additional incorporation of syntactic information in future work.

Table 1: Ablation study results and comparison with the state of the art for sentence compression generation in terms of F1 score and compression ratio. Also shown is the number of model parameters. We show that our model, *without additional syntactic information* as was used in previous methods, still obtains SOTA performance.

|  | Method | F1 | CR | Params |
|---|---|---|---|---|
| Zhao et al (2018) | LSTM implementation | 84.8 | 0.40 | — |
|  | Evaluator LM | 85.0 | 0.41 | — |
|  | Syntax-Based Evaluator LM | **85.1** | 0.39 | — |
| This paper | Baseline (BiLSTM) | 81.3 | 0.39 | 12M |
|  | No Fixation | 83.4 | 0.38 | 129M |
|  | Random TSM Init | 83.7 | 0.38 | 178M |
|  | TSM Weight Swap | 83.8 | 0.38 | 178M |
|  | Frozen TSM | 83.9 | 0.37 | 178M |
|  | Ours | **85.0** | 0.39 | 178M |

## 2 Ablation Study – Attention Maps

To shed more light onto the adapted TSM predictions for the conditions in our ablation study, we present saliency and neural attention maps for the conditions *Random TSM Init* and *TSM Weight*

Figure 1: Additional paraphrase generation attention maps from our ablation study, for both subnetworks (TSM predictions and upstream task attention) in our joint architecture. We show the TSM fixation predictions (left in blue) over epochs (last epoch is our converged models). We show the two-dimensional neural attention maps (right), showing the *Random TSM Init* (top) and *TSM Weight Swap* (bottom) model from our ablation study. The two-dimensional maps show the input sequence (horizontal axis) and the predicted sequence (vertical axis). We show the temporal TSM predictions over epochs, in order to depict how the fixation predictions change while training. The fixation predictions (for each epoch, left) are computed over words in the input sequences and then are integrated into the neural attention mechanism which in turn is used to make a prediction (vertical axis, right).

*Swap.* In Figure 1, we show that the adapted saliency predictions (blue, left showing) for paraphrase generation, between the two conditions (top vs. bottom) vary with respect to the words which are predicted to be most salient and the temporal adaptation during training. The last epoch is from the converged models, respectively. There exist notable differences in the adapted TSM predictions for the two ablations. However, we assume they do not play a role in performance between these two conditions, as these performance differences are not statistically significant. However, these conditions do perform significantly worse than our model (see paper for results). As shown in the paper, our model allocates the most attention to the word "travel" in the example sentence. This is the word that is changed in the paraphrase output, indicating that the our adapted TSM can effectively guide the paraphrase generation system. Figure 2 shows the adapted saliency predictions for the sentence compression task. The differences between both conditions are less distinct, with minimal temporal variation in the word saliency predictions. As with the paraphrase generation models, performance differences between the two ablations are not statistically significant. Compared to the saliency output for our model (shown in the paper), we observe that our model more equally allocates attention to the part of the sentence that is going to be deleted.

Figure 2: Additional sentence compression attention maps from our ablation study, for both sub-networks (TSM predictions and upstream task attention) in our joint architecture. We show the TSM fixation predictions (left in blue) over epochs (last epoch is our converged models). We show the two-dimensional neural attention maps (right), showing the *Random TSM Init* (top) and *TSM Weight Swap* (bottom) model from our ablation study. The two-dimensional maps show the input sequence (horizontal axis) and the predicted sequence (vertical axis). We show the temporal TSM predictions over epochs, in order to depict how the fixation predictions change while training. The fixation predictions (for each epoch, left) are computed over words in the input sequences and then are integrated into the neural attention mechanism which in turn is used to make a prediction (vertical axis, right).

While the 2d neural attention maps for the example sentence in the paraphrase generation task are similar for *Random TSM Init* and *TSM Weight Swap*, they differ clearly from the corresponding neural attention maps for our model (shown in the paper). Similarly, the 2d neural attention maps for sentence compression (Figure 2, right) are rather similar for *Random TSM Init* and *TSM Weight Swap*. However, the corresponding neural attention map for our method presented in the paper is more spread out and additionally allocates more attention on the position in the input sentence from which on the network decides to delete words. Taken together, these results illustrate the differences in neural attention that are connected to the superior performance of our full model over the ablation conditions.

## 3  Part of Speech Distributions – Content vs Function Words

In our paper we showed that our model and humans are significantly correlated with respect to gaze durations over part of speech tag (POS) distributions. We use this measure as POS tags have been shown to be good predictors of fixation probabilities [1, 3]. In Figure 3, we provide an

additional analysis on this matter. We group together the fixation duration predictions over content words (adjective, adverb, noun, and verb) and the fixation duration predictions over function words (conjunction, pronoun, determiner, numbers, adposition, and particles), for both human gaze and our model predictions (normalized between 0 to 1). In the figure, we show that our model predicts, *similarly to humans*, that content words are more informative than function words.

Figure 3: Per-sentence normalized gaze durations on content words versus function words for our TSM model and human gaze data across different corpora.

## 4 Links to Corpora

Here we provide the links for all publicly available corpora used. Note, due to licensing restrictions, the Dundee Corpus [2] is only available by directly contacting the authors.

```
https://www.quora.com/q/quoradata/First-Quora-Dataset-Release-Question-Pairs
https://github.com/google-research-datasets/sentence-compression
https://cs.nyu.edu/~kcho/DMQA/
https://osf.io/sjefs/
https://expsy.ugent.be/downloads/geco
https://www.perceptualui.org/publications/sood20_conll/
```