[Reviews · NeurIPS 2020]

Review 1

Summary and Contributions: Thank you for the author response. --- This paper addresses the problem of using eye-tracking data for NLP tasks. The paper presents two models. One is a text saliency model that learns fixation durations using syntactic data. The syntactic data is made based on a cognitive model of gaze during reading, E-Z reader model. The other model presented is a joint model that learns an NLP task using the output of the text saliency model in the attention layer. The joint model achieves state-of-the-art performances on the two different nlp tasks: paraphrasing and sentence compression. Also, ablation experiments show that the effectiveness of the proposed models.

Strengths: I think the work could be used for a variety of NLP applications since it proposes an approach to address the problem of lacking eye-tracking data (and I believe it is indeed a problem especially for some medical nlp applications). Also, it is interesting and valuable that the proposed model uses a cognitive theory to make syntactic eye-tracking data and shows they are effective.

Weaknesses: I don't see a major problem.

Correctness: The method and experiments seem ok. The claims are supported by the experiments.

Clarity: Yes. it is clearly written and I enjoyed reading it.

Relation to Prior Work: Yes. Explicitly using human attention for text comprehension tasks is new. Combining a cognitive theory and data-driven approaches is new. Combining gaze information in the attention layers for NLP is new.

Reproducibility: Yes

Additional Feedback: 1. Why did you combine BiLSTM and Transformer instead of just using transformer? The paper says "to better capture the sequential context". Did you try transformer only? 2. I'm wondering why there is no comparison with prior methods using eye gaze data. I can guess a few reasons, but could you give justification? 3. I'm wondering why the experiment section is in the current order -- joint model first. This is not critical, but just felt a bit strange because I expected evaluation of the text saliency model would come first. 4. typo: line 327, to coarse => too coarse?


Review 2

Summary and Contributions: The paper makes two contributions: 1) a way to bootstrap a reading text saliency model (TSM) from a cognitive model of reading gaze fixation and a small amount of fine-tuning data; 2) a way to incorporate and fine-tune the TSM in paraphrase and in sentence compression, showing that gaze prediction improves pure text models for those tasks.

Strengths: Nice, novel contributions, both in the TSM and in its incorporation in two reading comprehension tasks. Creative approach to build a strong TSM with very limited training data by using synthetic data from a cognitive model. Convincing empirical evaluation.

Weaknesses: The TSM architecture (GloVe>BiLSTM>Transformer) should be evaluated through ablation studies. For instance, a why not just a single (deeper) Transformer, maybe pre-trained as an MLM on a large text corpus to get good token embeddings? On the rebuttal, the authors mention preliminary experiments that supported their choice, I strongly encourage them to summarize those experiments in the final paper.

Correctness: Claims, method, and evaluation are solid and convincing.

Clarity: The paper was a real pleasure to read, only a couple of points where it could be improved for the reader. Packs a lot of information in an easy to digest way, I learned a lot from reviewing it.

Relation to Prior Work: As far as I can see, prior work is carefully addressed in a nicely organized Related Work section. I have not worked myself on these specific problems myself (gaze modeling and sentence paraphrase/compression) but I have colleagues who do, I follow their work closely, and I saw nothing of significance missing.

Reproducibility: Yes

Additional Feedback: 116-118: Did you compare BiLSTM+Transformer with Transformer alone to validate this? What about Transformer variants with longer contexts, such as Compressive Transformer? It would be good to be more precise about what is gained with the BiLSTM. Sec 3.2: Explain that h_i and s_j are computed according to task-specific architectures given in Sec 4, I was a bit lost here wondering about those architectural details. What’s the intuition behind the two score functions? Sec 4.1: This is where you should say what h_i and s_j mean for the two task networks. I can guess from the text, but it would be better to be explicit.


Review 3

Summary and Contributions: I like the paper in many ways, but it is based on a very fundamental, questionable premise: “Gaze has also been used to regularize neural attention layers via multi-task learning [3, 37]. To the best of our knowledge, however, no previous work has supervised NLP attention models by integrating human gaze predictions into neural attention layers.” This is, in fact, exactly what [3] does. In fact, their proposal is very related to the proposal here. While the proposal here multiplies in gaze-based attention weights, [3] use attention (over LSTM states) as a regularizer. Arguably, the second proposal models gaze and NLP tasks “more” jointly. I therefore suggest the authors instead focus on the real merits of their work: Using EZReader for pretraining. This is novel. It would also be interesting to see a more direct comparison of the two approaches to integrating gaze information. Such a comparison should ideally cover some/most tasks and set-ups used in previous work.

Strengths: The experiments all made sense, very thorough and present a substantial amount of work.

Weaknesses: Claims of novelty are false, at least in part, and relations with previous work not adequately discussed.

Correctness: Methods, yes, claims less so, I feel.

Clarity: Yes.

Relation to Prior Work: No, this is the main shortcoming.

Reproducibility: Yes

Additional Feedback:


Review 4

Summary and Contributions: Paper proposed an E-Z reader based pre-training objective to utilize human gaze information which can be potentially used to improve a downstream NLP task. Additionally, the paper discusses how the pre-trained model information can be injected into a task-specific network using attention mechanisms. Using empirical results on paraphrasing and sentence compression task, the paper claims that utilizing gaze information using a pre-trained model helps in achieving the state of the art results for both paraphrasing and sentence compression tasks. ### Update after reading author response ### I would like to thank authors for answering most of my questions/concerns. In particular, providing details about Li et al. [2019] experiment. I have revised my score accordingly.

Strengths: Paper has the following two novelties - Paper proposes a new pre-training technique based on E-Z Reader data which might be useful for many NLP tasks. - A simple yet intuitive way to add gaze information pre-trained knowledge to the task-specific DNN using attention The paper shows that their proposed TSM model's duration predictions are highly correlated with human gaze data which potentially shows that E-Z reader based pre-training is a good proxy to utilize gaze information in NLP models/tasks. Further, the paper provides an empirical evaluation using two public datasets for paraphrasing and sentence compression tasks claiming that such training helps in downstream tasks.

Weaknesses: There are multiple issues with the claims and evaluations presented in the paper. In particular, as a reader, I am not convinced that reported gains are due to exploiting gaze information. 1. An improvement over SOTA? : For paraphrasing task, the paper claims Patro et al. (2018) as SOTA which is an outdated baseline. [Decom_para ACL19] is a better baseline for comparison. Given that "No Fixation" method gives 27.81 BLEU-4 score with 69M params, I doubt that the proposed model's 28.82 BLEU-4 score with 79M is truly better than Patro et al. (2018)'s model. Ideally, authors should report the performance of baseline models using the same number of parameters. Similarly, on sentence compression task they should use a baseline with similar model params. With the current evaluation setup, it's not clear if gains can be attributed to the higher model capacity. 2. Evaluation: Paper reports only BLEU-4 scores for paraphrase task. Often people report multiple metrics to compare methods as a 1 point improvement in BLEU (27.81->28.82) on a single data might not mean anything in general. Usually, people report other metrics such as METEOR, ROUGE along with BLEU for a fair evaluation. For future revision of the paper, authors can also consider using more accurate metrics such as [BERTScore ICLR20], [BLEURT ACL20]. 3. Model architecture choice: What is the motivation of adding a transformer layer after a bilstm in text saliency model? The paper claims that this architecture allows us to better capture the sequential context without quantifying what do they mean by "better" ? Bi-LSTM followed by n-layer transformers in a non-standard NLP architecture so authors should describe what advantages does it provide over a standard Bi-lstm or a started transformer model? 4. Impact of pre-training on CNN and Daily Mail: Since the proposed models were pre-trained on CNN and Daily Mail and the baseline models are not pre-trained, it's not clear if the gains are due to model exploiting gaz information. We know that pre-training models on unlabeled corpus lead to better generalization performance across NLP tasks. I am still not convinced that predicting fixation durations provides any advantage over standard pre-training task such as masked language modeling. 5. Task/Dataset Choice: I think text summarization might be a good candidate to show the advantage of adding gaze information. Is there any particular reason for not considering that task? Also, to ensure that these techniques generalize, it's important to report numbers on more than 1 dataset for a given task. 6. Missing important implementation details: For the seq2seq model, author mentioned that they used greedy search. Is there any reason for not using a standard beam-search?

Correctness: Yes, the proposed method and evaluations are correct.

Clarity: Yes. I was able to understand most of the paper without any issues.

Relation to Prior Work: Paper discussed prior work related to exploiting gaze attention however doesn't discuss paraphrase models proposed in 2019 such as [Decom_para ACL19]. Refer to the Additional feedback section for missing references. Paper identifies it's main contribution in-terms of adding gaze information using attention.

Reproducibility: No

Additional Feedback: Please refer to weaknesses section for suggestions/questions. References: 1. [Decom_para ACL19] Decomposable Neural Paraphrase Generation https://arxiv.org/abs/1906.09741 2. [BERTScore ICLR20] BERTScore: Evaluating Text Generation with BERT https://arxiv.org/abs/1904.09675 3. [BLEURT ACL20] BLEURT: Learning Robust Metrics for Text Generation https://arxiv.org/abs/2004.04696

[Author Response · NeurIPS 2020]

We thank all reviewers for their detailed and valuable feedback. In the following, we address the reviewers' comments and questions, which we will clarify in the final revision.

**Novelty of the joint model [R3]** We agree with R3 that [3,37] pioneered gaze integration in NLP tasks, paving the way for future works like ours. We did not intend to claim we are the first to propose gaze integration in NLP. Instead, a key novelty of our work is our joint modelling approach to attention and comprehension. Instead of only using gaze information as additional supervision during training, we propose a joint training framework in which our pre-trained text saliency model (TSM) is adapted to a target task for which no gaze data is available. We will clarify in the paper that this joint training framework and not the specific method used for gaze integration is one of our core contributions.

**Comparison to prior methods using gaze data in NLP [R1,R3]** A core contribution of our work is the joint modelling of attention and comprehension for text comprehension tasks, which we show to improve performance in extensive evaluations. We value the suggestion made by R1 and R3 to additionally compare our approach to previous works using gaze in NLP tasks ([3] and [37]). When comparing our results for sentence compression on the google dataset to [37], who employ gaze as an additional supervision in a multi-task learning setup, we observe an increase of ~5% F1 score for our method. A comparison to [3] is outside the scope of our current work as the focus of [3] is on sequence classification, while ours is on modeling human attention with text comprehension (paraphrasing and summarization). To the best of our knowledge, no previous works studied gaze integration for the paraphrase generation task.

**Evaluation to Current SOTA + Baseline Models [R4]** We agree with [R4] that an evaluation against the recent work on paraphrase generation by Li et al. [2019] would be desirable. Unfortunately, several works, including Li et al. [2019] cite the Kaggle Competition release of the Quora Question Pairs (QQP) dataset and use an undocumented split and subset of the original QQP dataset. The authors did not reply to our requests for details on the splits. Therefore, we compared to the state-of-the-art for which a reproducible split was published, citing the original QQP dataset and using the same splitting protocol, to ensure comparability [24,54]. When comparing our BLEU-4 scores (28.82 BLEU-4) to Li et al. [2019] (best BLEU of 25.03), in spite of the unknown dataset split, we still outperform this recent work.

Regarding the number of parameters in the baseline models, we tried to reach the authors of [54,85] to obtain this information, but they did not get back to us. Crucially, we show the effectiveness of our joint training approach cannot be explained by an increase in parameters as we compare our full model against ablated variants with the same number of parameters (*Random TSM Init*, *TSM Weight Swap*, and *Frozen TSM*). All ablations are inferior to our full model.

We use BLEU because it is an established metric [37] which is currently still agreed on as a standard for text generation tasks [53]. We also show that our improvements in BLEU-4 score are statistically significant using paired t-tests. We agree that additional metrics can provide more information regarding the generated text. We welcome the reviewers suggestion on the novel metrics published at ICLR20 and ACL20, less than two months before the NeurIPS deadline and will add all suggested metrics to the revised version, providing additional points of comparison for future work.

**Model Architecture Choices [R1,R2,R4]** We chose a BiLSTM with a Transformer for our TSM model, as this combination yielded predictions most similar to human data in preliminary experiments, particularly with longer sequence lengths. According to Wang et al. [2019], this might be due to the transformer encoding coarse relational information about positions of sequence elements, while the BiLSTM better captures fine grained word level context. We will a detailed ablation study of the TSM in the supplementary material. Lastly, for decoding we use greedy search.

**Impact of Pre-training on CNN and Daily Mail [R4]** By pre-training on the CNN and Daily Mail corpora using EZ-Reader and subsequently fine-tuning on human gaze data, our TSM model learns to accurately predict human gaze on sentences. These human-like saliency weights output by the TSM are already beneficial to the upstream tasks without further joint training (Frozen TSM), and improve even more during the joint training process. R4 raises the question of whether e.g. a masked language model trained on the CNN and Daily Mail can lead to a comparable performance when integrated into the joint model. While we agree that this is an interesting research question, it lies outside the scope of this paper in which we investigate the impact of a text saliency model in our joint training framework.

**Task Choice + Generalizability [R4]** We chose sentence compression, which is a text summarization task. Studying extractive summarization alongside paraphrase generation allows us to show generalizability of our method across two tasks with different output formats, requiring different evaluation metrics (BLEU and F1 score). In our opinion, the positive results on both of these different tasks are a strong statement in favor of the generalizability of our approach.

# References

Zichao Li, Xin Jiang, Lifeng Shang, and Qun Liu. Decomposable neural paraphrase generation. In *ACL*, 2019.

Zhiwei Wang, Yao Ma, Zitao Liu, and Jiliang Tang. R-transformer: Recurrent neural network enhanced transformer. *arXiv:1907.05572*, 2019.


[Meta-Review · NeurIPS 2020]

Overall, the reviewers appreciated this paper and thought it was an interesting contribution to the literature. Because of this I am recommending that the paper be accepted. However, there was one major request from the Reviewer 3 (and me) that the authors appropriately frame their discussion of previous work, specifically [3]. Quoting reviewer 3's discussion directly: "The authors claim to be first to directly supervise attention; which has been done before with token-level annotation, and is exactly what [3] does with gaze. This false claim of novelty is problematic, but also unnecessary, since this is a great paper that already makes decent contributions, eg smart pretraining." I would highly encourage the authors to take this comment in earnest, and give the relationship with [3] proper treatment and explaining the additional contributions that this paper makes on top of it.